# Inhibition of Osteoclastogenesis by Thioredoxin-Interacting Protein-Derived Peptide (TN13)

**DOI:** 10.3390/jcm8040431

**Published:** 2019-03-29

**Authors:** Mi Jeong Kim, Won Sam Kim, Jae-Eun Byun, Jung Ha Choi, Suk Ran Yoon, Inpyo Choi, Haiyoung Jung

**Affiliations:** 1Immunotherapy Research Center, Korea Research Institute of Bioscience and Biotechnology (KRIBB), Yuseong-gu, Daejeon 34141, Korea; tito006@naver.com (M.J.K.); kwasa@ensolbio.co.kr (W.S.K.); quswodms@kribb.re.kr (J.-E.B.); cjhsong5284@kribb.re.kr (J.H.C.); 2Department of Biochemistry, School of Life Sciences, Chungbuk National University, Cheongju 28644, Korea; 3Department of Functional Genomics, University of Science and Technology, Yuseong-gu, Daejeon 34113, Korea

**Keywords:** osteoporosis, osteoclast, osteoclastogenesis, p38 MAPK, TAT-TN13, ovariectomy

## Abstract

Overactivated osteoclasts lead to many bone diseases, including osteoporosis and rheumatoid arthritis. The p38 MAPK (p38) is an essential regulator of the receptor activator of nuclear factor-κB ligand (RANKL)-mediated osteoclastogenesis and bone loss. We previously reported TAT conjugated thioredoxin-interacting protein-derived peptide (TAT-TN13) as an inhibitor of p38 in hematopoietic stem cells (HSCs). Here, we examined the role of TAT-TN13 in the differentiation and function of osteoclasts. TAT-TN13 significantly suppressed RANKL-mediated differentiation of RAW 264.7 cells and bone marrow macrophages (BMMs) into osteoclasts. TAT-TN13 also inhibited the RANKL-induced activation of NF-κB and nuclear factor of activated T-cells cytoplasmic 1 (NFATc1), leading to the decreased expression of osteoclast-specific genes, including tartrate-resistant acid phosphatase (TRAP) and Cathepsin K. Additionally, TAT-TN13 treatment protected bone loss in ovariectomized (OVX) mice. Taken together, these results suggest that TAT-TN13 inhibits osteoclast differentiation by regulating the p38 and NF-κB signaling pathway; thus, it may be a useful agent for preventing or treating osteoporosis.

## 1. Introduction

Osteoporosis is the most common metabolic bone disease caused by decreased bone mass per unit volume and micro architectural deterioration of bone tissue, which results in weakened bone strength; thus, bones can easily get fractured [1]. Osteoporosis is classified into two types: primary and secondary osteoporosis. The main cause of primary osteoporosis is bone loss that usually occurs in women after menopause and in both men and women during the normal aging process. Secondary osteoporosis, is directly caused by bone loss that results from a disease or medication. Throughout one’s lifetime, bone undergoes a dynamic process termed remodeling which is regulated by a balance between bone formation by mesenchymal stem cells-derived osteoblasts, and bone resorption by osteoclasts, which are derived from monocyte–macrophage lineage of the hematopoietic cells. However, excessive bone resorption by osteoclasts can lead to osteolytic diseases including osteoporosis, periodontal disease, rheumatoid arthritis, and bone metastatic disease [2,3]. Osteoclasts are the main targets of therapeutics for the prevention and treatment of such diseases. They are tissue-specific macrophage polykaryons, which originate from the hematopoietic progenitors of the monocyte/macrophage lineage at or near the bone surface [4].

The receptor activator of nuclear factor-κB ligand (RANKL) and the macrophage monocyte colony-stimulating factor (M-CSF) are essential signaling molecules for osteoclast differentiation [4,5]. RANKL is a member of the tumor necrosis factor (TNF) superfamily and is produced by osteoblasts and bone marrow stromal cells. The conjunct of RANKL and RANK on osteoclast cell surface recruits signaling adapter molecules, such as TNF receptor-associated factor-κB ligand 6 (TRAF6) which subsequently associates with transforming growth factor beta-activated kinase 1 (TAK1) activates downstream signaling pathways such as mitogen-activated protein kinases (MAPKs), AKT and c-jun N-terminal kinase (JNK), and thereby induces the expression of key transcription factors and osteoclast-specific genes, including TRAP, Cathepsin K, matrix metalloproteinase 9 (MMP-9), and C-Src [4,6].

p38 MAPK (p38) is activated by many cytokines, growth factors and important regulators of bone remodeling such as RANKL, transforming growth factor β-1 (TGFβ-1), bone morphogenetic protein (BMP) and TNF-α. RANKL induces the activation of p38 pathway by interacting with receptor of activated protein C kinase 1 (RACK1), TAK1 and MAP kinase kinase 3/6 (MKK3/6). TAK1 and MKK3/6 have been reported to play positive roles in proliferation, differentiation, and/or survival of osteoclasts [7]. In addition, MKK3 and MMK6 deficient mice have shown to exhibit a higher trabecular bone density as compared to control littermates at 4 months of age, and despite normal bone formation, the number of osteoclasts was decreased. Additionally, it was shown that MKK3 and MKK6 deficient ovariectomized (OVX) mice were partially protected from bone loss caused by estrogen withdrawal due to decreased bone resorption [8]. Osteoclast differentiation by macrophage colony-stimulating factor (M-CSF) and RANKL has been shown to be suppressed in MKK3 or MKK6 deficient bone marrow cells due to reduced p38 activation and inhibited expression of NFATc1 and other osteoclast-specific markers, such as Cathepsin K, osteoclast-associated receptor (OSCAR) and MMP-9, leading to an impaired bone resorption [8,9,10]. In mammalian cells, there are four isoforms of p38: p38α, -β, -γ, and -δ [11,12]. Among these isoforms, p38α is the only one being highly expressed in osteoclast precursors and mature osteoclasts.

In addition, p38α deficient bone marrow hematopoietic cells were not able to differentiate into mature osteoclasts in response to M-CSF and RANKL treatment in vitro [13]. Consistent with these data, post-developmental deletion of p38α using Mx-cre-mediated conditional gene inactivation resulted in decreased number of osteoclasts, reduced bone resorption, and increased bone density under physiological conditions. Moreover, deletion of p38α has been shown to protect mice against TNF-α-mediated arthritis and systemic bone loss [13]. In our previous studies have shown that thioredoxin-interacting protein (TXNIP) interacted with p38α via docking interaction and inhibited p38 activity in HSCs. Based on these results, we developed a 13-amino-acid-containing peptide (TN13) from TXNIP docking motif to target p38α. To deliver the synthesized peptide into cells, we conjugated a HIV TAT transduction domain sequence (YGRKKRRQRRR) to the N-terminus of the TN13 peptide (TAT-TN13), which efficiently penetrated HSCs and inhibited p38 activity [14,15].

In this study, given the importance of p38 signaling pathway in osteoclastogenesis, we hypothesized that TAT-TN13 might represent a novel treatment for osteoclast-related diseases. We investigated the effects of TAT-TN13 on RANKL-induced osteoclastogenesis and evaluated its efficacy in vivo using ovariectomized mouse as a model of osteoporosis. We showed that TAT-TN13 inhibited RANKL-stimulated osteoclastogenesis and bone resorption by suppressing p38/NF-κB/NFATc1 signaling pathways, which regulated gene expression of TRAP and Cathepsin K. In addition, TAT-TN13 reduced bone loss in OVX mice. Taken together, these data suggest that TAT-TN13 may be a potential therapeutic drug for treating osteoporosis and bone diseases caused by exacerbated osteoclast formation.

## 2. Experimental Section

### 2.1. Mice

C57BL/6 female mice (aged 8 weeks) were obtained from The Jackson Laboratories (Bar Harbor, ME, USA). All mice were housed in a pathogen-free animal facility under with 12 h light/dark cycle at 23 °C. Animals were maintained in accordance with the Guide for the Care and Use of Laboratory Animals published by the US National Institutes of Health and were handled according to protocols approved by the Korea Research Institute of Bioscience and Biotechnology.

### 2.2. Cells, Media, and Reagents

Murine RAW 264.7 monocytic cell line was obtained from American Type Culture Collection (Rockville, MD, USA). The alpha modification of Eagle’s medium (α-MEM) and Dulbecco’s Modified Eagle’s Medium (DMEM) were purchased from WELGENE (WELGENE Inc, Gyeongsan, Korea). Penicillin/streptomycin and fetal bovine serum (FBS) were obtained from Gibco-BRL (Gaithersburg, MD, USA). Macrophage colony stimulating factor (M-CSF) and recombinant murine sRANK ligand were procured from PeproTech (Rocky Hill, NJ, USA). TRAP staining was performed using a leukocyte acid phosphatase kit (387-A) from Sigma (St. Louis, MO, USA) and p38 inhibitor SB203580 was purchased from Calbiochem Corp (La Jolla, CA, USA).

### 2.3. Cell Culture

Primary osteoclasts were derived from bone marrow monocytes (BMMs). Bone marrow monocytes (BMMs) were isolated from the femoral bone marrow of 8-week-old C57BL/6 mice. Briefly, BMMs were cultured in α-MEM containing 10% FBS, 100 U/mL penicillin, 100 mg/mL streptomycin and M-CSF (10 ng/mL) for 16–24 h in a humidified atmosphere with 5% CO_2_ at 37 °C. After 24 h, the non-adherent osteoclast precursor cells were resuspended in 100-mm dishes with 30 ng/mL of M-CSF and were allowed to adhere for 3 days. After BMMs were cultured for 3 days, cells were detached from dish surface using 0.25% trypsin-EDTA. Cells were then cultured in 24-well plates at a density of 4.2 × 10^3^ cells/cm^2^ in α-MEM containing RANKL (50 ng/mL) and M-CSF (30 ng/mL) for 7 days. Simultaneously, RAW 264.7 cells were seeded in 12-well plates at a density of 1.32 × 10^4^ cells/cm^2^ in α-MEM containing 10% FBS, 100 U/mL penicillin, 100 mg/mL streptomycin and RANKL (40 ng/mL). Cells were then incubated for 4 days in a humidified atmosphere with 5% CO_2_ at 37 °C. In order to check the effects of TAT-TN13 on the osteoclastogenesis, cells were treated with SB203580 (10 µM) and TAT-TN13 (0, 10, or 20 µM) every day and culture medium was replaced every 2 days.

### 2.4. Cell Viability Assay

To determine the effect of TAT-TN13 peptide on the cell viability, cytotoxicity assays were performed using the Cell Counting Kit-8 (Dojindo, Kumamoto, Japan) and trypan blue exclusion assay as described previously [16]. RAW 264.7 cells were seeded in 96-well plate at a density of 6.25 × 10^4^ cells/cm^2^. After attachment, the cells were treated with varying concentrations of TAT-TN13 peptide for 24 h or 4 days. The reagent (from CCK-8) was added to each well, and incubating at 37 °C for 1 h. Optical density was measured at 450 nm by a microplate reader (SpectraMax^®^i3, Molecular Devices, San Jose, CA, USA). For trypan blue exclusion assay RAW 264.7 cells were seeded in 6-well plate at a density of 5.3 × 10^4^ cells/cm^2^. After attachment, the cells were treated with varying concentrations of TAT-TN13 peptide for 24 h. After treatment for 24 h, cells were then harvest and resuspended in 1 mL of phosphate-buffered saline (PBS). Subsequently, 10 µL of cell suspension was then mixed with 10 µL of 0.4% Trypan blue and the mixture was incubated for 3 min; 10 µL of cell suspension was loaded into the hemocytometer and cells were counted under a light microscope.

### 2.5. TRAP Staining

Tartrate-resistant acid phosphatase (TRAP) activity, a histochemical marker of osteoclasts, was stained using the Acid Phosphatase, Leukocyte TRAP Kit (Sigma-Aldrich, St. Louis, MO, USA) according to the manufacturer’s protocol. Adherent cells were fixed with 4% paraformaldehyde for 1 min and washed using saline. Subsequently, cells were treated with ethanol/acetone (50:50 v/v) for 1 min, air dried and incubated for 10 min at room temperature in an acetate buffer (0.1 M sodium acetate, pH 5.0) containing 0.01% naphthol AS-MX phosphate (Sigma, St. Louis, MO, USA) and 0.03% fast red violet LB salt (Sigma, St. Louis, MO, USA) in the presence of 50 mM sodium tartrate. Stained cells were observed under microscope, and TRAP-positive multinucleated cells (≥3 nuclei) were counted as osteoclasts.

### 2.6. Resorption Pit Assay by Osteoclasts

RAW 264.7 cells were seeded at a density of 1.05 × 10^4^ cells/cm^2^ in Corning Osteo Assay Surface 24-well plates coated with calcium phosphate substrate. After attachment, cells were treated with 40 ng/mL RANKL and TAT-TN13 (10 or 20 µM) until the formation of mature osteoclasts. The medium was replaced with fresh media every 2 days. After the incubation at 37 °C for 4 days, cells were washed with 10% bleach solution to remove the adherent osteoclasts. Cells were then rinsed with PBS and air-dried for better visualization of the surface. Images of resorption pits on the plates were visualized under a scanning electron microscope (Olympus IX53; Olympus, Japan), and the bone resorption area was quantified using Olympus CellSens imaging software.

### 2.7. RNA Isolation and Quantitative Real-Time PCR

Total RNA was extracted from treated cells by using RNeasy Mini Kit (Qiagen, Hilden, Germany) according to the manufacturer’s instructions. About 1 µg of total RNAs were reverse-transcribed to first strand cDNAs by using ReverTra Ace® qPCR RT Kit (Toyobo, Osaka, Japan) and analyzed by real time PCR (Takara Bio, Kusatsu, Japan) using SYBR Green PCR Master Mix (Takara Bio, Kusatsu, Japan) with specific primers. The mRNA expression level was calculated using GAPDH as a control. The primer sequences were as follows: GAPDH, 5′-CTGCGACTTCAACAGCAACT-3′ and 5′-GAGTTGGGATAGGGC CTCTC-3′; NFATc1, 5′-CTCGAAAGACAGCACTGGAGCAT-3′ and 5′-CGGCTGCCTTCC GTCTCATAG-3′; TRAP, 5′-TCCTGGCTCAAAAAGCAGTT-3′ and 5′-ACATAGCCCACA CCGTTCTC-3′.

### 2.8. Luciferase Reporter Assay

To examine the inhibitory effects of TAT-TN13 on NF-κB transcriptional activities, RAW 264.7 cells were transiently co-transfected with pNF-κB–Luc (Stratagene, La Jolla, CA, USA) and pRL-CMV reporter vectors (Promega, Madison, CA, USA) using Lipofectamine and Plus reagent (Invitrogen, CA, USA). Briefly, 2 × 10^5^ cells were plated in 6-well plates and pretreated with TAT-TN13 and the p38 inhibitor for 1 h and then stimulated with 40 ng/mL of RANKL in the presence for 8 h. Cells were then harvested and lysed in a reporter lysis buffer (Promega, Madison, WI, USA). Luciferase assays were performed according to instructions supplied with the luciferase assay kit (Promega, Madison, WI, USA). Firefly luciferase and Renilla luciferase activities were measured by a microplate reader (SpectraMax^®^i3, Molecular Devices, San Jose, CA, USA).

### 2.9. Western Blot Analysis

For Western blotting analysis, cultured cells were washed twice with PBS and subsequently lysed with RIPA I buffer (25 mM HEPES, pH 7.7, 0.3 M NaCl, 1.5 mM MgCl2, 0.2 mM EDTA, 0.1% Triton X-100, 10 mM β-glycerophosphate, 1 mM NaF, 1 mM Na_3_VO_4_) containing protease inhibitor cocktail (Roche Applied Science, Basel, Switzerland). Protein concentration was estimated in whole-cell lysates, lysates with the same amount of protein content were separated by sodium dodecyl sulfate polyacrylamide gel electrophoresis (SDS-PAGE; BioRad, Hercules, CA, USA), followed by their transfer onto a PVDF membrane (Millipore, Bedford, MA, USA) by immunoblotting. The membrane was then probed with anti-phospho-p38 (Cell Signaling Technology, Danvers, MA, USA), anti-p38 (Cell Signaling Technology, Danvers, MA, USA), anti-phospho-p65 (Cell Signaling Technology, Danvers, MA, USA), anti-p65 (Cell Signaling Technology, Danvers, MA, USA), anti-NFATc1 (Santa Cruz Biotechnology, Dallas, CA, USA), anti-Cathepsin K (Santa Cruz Biotechnology, Dallas, CA, USA), anti-TRAP (Santa Cruz Biotechnology, Dallas, CA, USA), anti-PU.1 (Santa Cruz Biotechnology, Dallas, CA, USA) and anti-β-actin (Santa Cruz Biotechnology, Dallas, CA, USA) antibodies, followed by incubation with HRP-conjugated secondary antibody. Protein bands were then detected using ECL kit and Western blots were imaged using WSE-6100 LuminoGraph (ATTO, Tokyo, Japan) as described previously [17].

### 2.10. Immunofluorescence Analysis of NF-κB

RAW 264.7 cells were plated on cover slips in 12-well plates for 24 h. After incubation, the cells were treated with or without 20 µM TAT-TN13 in the presence of RANKL (40 ng/mL) for 1 h. After incubation, cells were washed with PBS, fixed with 4% paraformaldehyde in PBS for 20 min at room temperature, followed by permeabilization with 0.2% Triton X-100 for 15 min at 4 °C. Cells were then blocked with 5% BSA at 37 °C for 1 h and incubated with rabbit anti-p65 antibody (Cell Signaling Technology, Danvers, MA, USA) overnight at 4 °C. Cells were then washed with PBS, and incubated with Alexa Fluor 488-conjugated secondary antibody (Life technology, Waltham, MA, USA) for 1 h at room temperature. Finally, cells were washed 2 times with PBS, and then were mounted with DAPI containing mounting reagent (Invitrogen, Carlsbad, CA, USA). The images were captured using LSM510 confocal microscope (Carl Zeiss, Gottingen, Germany).

### 2.11. Ovariectomized (OVX) Mouse Model

Eight-week-old C57BL/6 female mice were generally anesthetized for surgery with 1.5–4% isoflurane in oxygen and subjected to either a sham operation or bilateral OVX. We randomly divided the mice into three groups (*n* = 6/group): a sham treated group, ovariectomized (OVX) mice treated with normal saline, and OVX mice treated with TAT-TN13 dissolved in normal saline. After 1 week, intraperitoneal injection was performed every other day for 6 weeks as follows: the sham and OVX groups were injected with saline and TAT-TN13 groups were injected with 20 mg/kg TAT-TN13. Mice were sacrificed after 6 weeks and their left femur was removed and placed in 1.5 mL tubes containing 4% paraformaldehyde. Micro-CT (Quantum FX, Perkin Elmer, Waltham, MA, USA) was performed to detect femur metaphysis and obtain bone mineral density (BMD), trabecular number (Tb.N), bone surface (BS), bone surface per total volume (BS/TV), bone volume/tissue volume ratio (BV/TV), trabecular thickness (Tb.Th) and trabecular spacing (Tb.Sp).

### 2.12. Histological Analysis

Mice femurs were fixed in 4% formalin, decalcified for 2 weeks using 10% tetrasodium-EDTA aqueous solution and embedded in paraffin. Sections (4 µm thick) were prepared using microtome (Jung, Heidelberg, Germany). Histological sections were stained with TRAP and hematoxylin-eosin (H&E) and evaluated by microscope (Olympus IX53; Tokyo, Japan).

### 2.13. Statistical Analysis

Data collected from each (experimental and control) group were expressed as mean ± SD (standard deviation). Statistical analyses were analysed using SPSS (Statistical Package for Social Science, SPSS Inc., Chicago, IL, USA) 19.0 software (IBM). ANOVA with Tukey-Kramer’s post-hoc test was used for comparisons between the groups. * *p* < 0.05, ** *p* < 0.01 and *** *p* < 0.001 were considered to be statistically significant.

## 3. Results

### 3.1. TAT-TN13 Inhibits RANKL-Induced Osteoclast Differentiation

In order to select optimal nontoxic concentrations of TAT-TN13 in this study, we examined viability of RAW264.7 cells exposed to different concentrations of TAT-TN13 using trypan blue dye exclusion assay and CCK-8 assay. After 24 h of treatment with TAT-TN13, it was observed that cell viability was not changed at doses below 20 µM, whereas at 50 and 100 µM, it reduced significantly. 50% lethal concentration (LC50) of TAT-TN13 in RAW264.7 cells was 86.04 µM (Figure 1A,B). We also tested long-term cytotoxicity of TAT-TN13 with or without RANKL for 4 days in RAW 264.7 cells. Cell viability was not changed at doses below 20 µM of TAT-TN13 treatment (Appendix A). Therefore, TAT-TN13 concentrations below 20 µM were used for further experiments. Next, we investigated whether TAT-TN13 suppressed RANKL-induced p38 activity. The phosphorylation of p38 was induced by RANKL treatment and was markedly suppressed by TAT-TN13 treatment in RAW 264.7 cells (Figure 1C,D).

To examine the effects of TAT-TN13 on osteoclastogenesis, RAW 264.7 cells were differentiated into osteoclasts by RANKL treatment in the absence or presence TAT-TN13 (10 or 20 µM) or SB203580 (10 µM), a p38 MAPK pathway inhibitor [9,18]. We then performed TRAP staining to visualize TRAP-positive multinucleated osteoclasts by light microphotography. The osteoclast formation induced by RANKL was strongly inhibited by TAT-TN13 treatment as in SB203580 treated RAW 264.7 cells (Figure 2A). The inhibitory effect of TAT-TN13 against RANKL-stimulated osteoclast differentiation in RAW 264.7 cells were evaluated by the counting the number of TRAP-positive multinucleated osteoclasts. TAT-TN13 significantly reduced the number of multinucleated TRAP-positive cells in a dose-dependent manner as in SB203580. At 20 µM concentration, TAT-TN13 was observed to inhibit osteoclast formation by approximately 90% as compared to control cells (Figure 2B). To investigate whether TAT-TN13 inhibited osteoclast formation of primary bone marrow macrophages (BMMs), we treated BMMs with RANKL and M-CSF in the presence of TAT-TN13. TAT-TN13 also significantly inhibited the osteoclast formation of BMMs (Figure 2C). These data showed that TAT-TN13 efficiently inhibited RANKL-induced osteoclast differentiation of primary BMMs and RAW 264.7 cells in vitro.

### 3.2. TAT-TN13 Reduces the Number of Mature Osteoclasts

TAT-TN13-treated cells were shown to reduce the area of osteoclast bone resorption pits in a dose-dependent manner (Figure 3A,B). To determine whether TAT-TN13 affected the bone resorbing activity of osteoclasts, we fully differentiated RAW 264.7 cells into osteoclasts for 4 days and then treated TAT-TN13 or SB203580 for 12 h. The regulation of p38 signaling pathways by TAT-TN13 or SB203580 treatment inhibited osteoclastogenesis and it did not affect for the bone resorbing activity of mature osteoclasts (Appendix A). These results suggested that TAT-TN13 inhibited osteoclastogenesis in RAW 264.7 cells and did not inhibit the bone resorbing activity of mature osteoclasts in vitro.

### 3.3. TAT-TN13 Suppresses RANKL-Induced Osteoclast-Related Genes Expression

To further assess the inhibitory effects of TAT-TN13 on osteoclastogenesis in response to RANKL, we examined the expression of osteoclast-related genes, including *TRAP* and *NFATc1*. As shown in Figure 4A, B, TAT-TN13 significantly inhibited the expression of osteoclast-related genes. These results coincided with the inhibitory effect of TAT-TN13 on the number of mature osteoclasts presented in Figure 3. Altogether, these data demonstrated that TAT-TN13 inhibited osteoclastogenesis by suppressing the expression of osteoclast-related genes in vitro. p38 is known to activate several downstream pathways; especially, the NF-κB pathway has been proven to play an essential role in the regulation of RANKL-induced osteoclastogenesis [19]. To determine the underlying mechanism associated with TAT-TN13 treatment, we evaluated the inhibitory effects of TAT-TN13 on RANKL-induced transcriptional activity of NF-κB using a luciferase activity assay. RANKL significantly increased the transcriptional activity of NF-κB in RAW 264.7 cells that were transiently transfected with NF-κB-luc, whereas treatment with TAT-TN13 or SB203580 effectively reduced this enhanced activity (Figure 4C). These results suggested that TAT-TN13 inhibited the RANKL-induced NF-κB signaling pathway via inhibiting p38 activity, and thereby suppressed osteoclast formation.

### 3.4. TAT-TN13 Suppresses RANKL-Induced NF-κB Activation

NF-κB plays an important role in the induction of NFATc1 in RANKL-induced osteoclastogenesis [20]. Therefore, we investigated the effect of TAT-TN13 on the activation of NF-κB and NFATc1 induction. RANKL stimulation led to the phosphorylation of the NF-κB p65 subunit and induction of NFATc1 expression, which were significantly inhibited by TAT-TN13 treatment in RAW 264.7 cells (Figure 5A). Under unstimulated conditions, NF-κB is predominantly present in an inactive form in the cytoplasm through its association with IκB proteins, which are inhibitors of NF-κB. Upon stimulation with signaling molecules, NF-κB is released from IκB and translocated to the nucleus where it regulates gene transcription [21]. Using confocal microscopy, we found that TAT-TN13 treatment inhibited RANKL-induced nuclear translocation of the p65 subunit of NF-κB in RAW 264.7 cells (Figure 5B). In addition, we investigated the expression of signaling molecules which were considered to be involved in osteoclast differentiation and activation in RAW 264.7 cells treated with RANKL for 0, 1, 2 or 3 days. As expected, RANKL significantly increased the phosphorylation of p38, p65 and expression of Cathepsin K and TRAP during the osteoclastogenesis. In contrast, PU.1 was time-dependently decreased (Figure 5C). Subsequently, we evaluated whether TAT-TN13 could regulate the expression of genes that encode proteins related to osteoclast differentiation and activation in RAW 264.7 cells. The expression levels of osteoclast-related genes were markedly decreased by TAT-TN13 treatment in a dose-dependent manner (Figure 5D). Thus, our findings demonstrated that TAT-TN13 suppressed RANKL-induced osteoclastogenesis by inhibiting NF-κB and NFATc1 pathways.

### 3.5. TAT-TN13 Prevented OVX-Induced Bone Loss In Vivo

To demonstrate the therapeutic effects of TAT-TN13 in vivo, we intraperitoneally injected TAT-TN13 into OVX-induced osteoporotic mice. Among various methods of analysis, x-ray micro-computed tomography (µCT) was used for two- and three-dimensional analysis of the changes in the femoral bone. Seven weeks after operation, successful OVX-induced osteoporosis was confirmed from μCT 3D images. Representative μCT images of the distal femur showed that the trabecular pattern of vehicle-treated OVX group was relatively vacant as compared to the other groups, whereas in TAT-TN13-treated OVX group trabecular bone volume was higher compared to vehicle-treated OVX group (Figure 6A). For the quantitative analysis, bone mineral density (BMD, mg/cm^3^), trabecular number (Tb.N, mm−1), bone surface density (BS, mm^2^), bone surface density /trabecular volume (BS/TV, mm^2^ mm^3^), bone volume/trabecular volume (BV/TV, %), trabecular thickness (Tb.Th, mm) and trabecular separation (Tb.Sp, mm) were determined. As shown in Figure 6B–H, the TAT-TN13-treated OVX group exhibited significantly lesser bone loss than the vehicle-treated OVX group, without affecting thickness, or separation. As per μCT morphometric analyses, the values for these parameters were significantly higher in TAT-TN13-treated OVX group compared to those in vehicle-treated OVX group. The vehicle-treated OVX group was significantly weighed more than sham or TAT-TN13-treated OVX group following ovariectomy (Figure 6I). Besides, TRAP staining and histomorphometric analysis of sections of femurs revealed that there was substantial reduction in both osteolysis and osteoclast surface in bone samples of TAT-TN13-treated OVX group in comparison with vehicle-treated OVX group (Figure 6J,K). These results are consistent with the in vitro results and indicate that TAT-TN13 is an efficient agent for inhibiting OVX-induced bone loss in vivo.

## 4. Discussion

Osteoclasts differentiate from hematopoietic precursors of monocyte/macrophage lineage and are involved in bone morphogenesis, remodeling, and resorption. The enhanced formation or activity of osteoclasts is the major cause of diseases related to excessive bone resorption, including wear particle-induced osteolysis, osteoporosis, rheumatoid arthritis, multiple myeloma, and metastatic cancers [4,22,23]. Therefore, in this study, we focused on the compounds that can potentially inhibit osteoclastogenesis in order to prevent or treat osteolytic diseases. Current treatments developed to target osteoclasts are estrogen-replacement therapy, bisphosphonates (BPs), and denosumab [24]. However, such therapies can increase the risk of adverse effects, including breast cancer, hypercalcemia, stroke, heart attack and hypertension. Although the humanized anti-RANKL neutralizing monoclonal antibody, denosumab, has been widely used as an efficient, safe, and cost-effective treatment for osteoclast-related diseases, its long-term efficacy has not been confirmed yet [25,26,27].

Osteoclast differentiation and activation require the activation of several MAPK pathways, amongst which the p38 family of MAPKs, most notably p38α, is an essential regulator of RANKL-mediated osteoclastogenesis [13]. Thus, it is a potential therapeutic target for osteoporosis. Interestingly, several studies have shown that the therapeutic use of synthetic small-molecule inhibitors of p38 activation prevented inflammation and bone destruction in experimental arthritis models and demonstrated clinical efficacy. In addition, several clinical trials have tested the efficacy of these compounds in human diseases [28,29,30,31,32]. However, concerns have been raised about severe side effects of general blockade of p38 family [33,34,35].

Previously, we found that TAT-TN13 was a potent inhibitor of p38α isoform with specificity higher than SB203580 in aged HSCs [14]. This led to our hypothesis that TAT-TN13 could inhibit RANKL-induced osteoclast differentiation. In this study, we demonstrated that TAT-TN13 inhibited RANKL-stimulated osteoclast differentiation by blocking the p38 signaling pathway. In addition, several downstream targets were identified in response to TAT-TN13 treatment to investigate the mechanisms through which it might affect osteoclastogenesis. RANKL binding to RANK has been shown to induce trimerization of TRAF6, resulting in activation of NF-κB and MAPK signaling [36]. It has been found that by phosphorylating the p65 subunit of NF-κB at Ser536, p38 contributes to RANKL-induced osteoclast formation, and thus, increases NF-κB transcriptional activity [10]. In addition, p38a has been shown to be a major regulator of NFATC1 and not only phosphorylates NFATc1 but also enhances nuclear accumulation of NFATc1 and of the transcriptional activation of the cathepsin K gene promoter [37]. Several studies have shown that NF-κB signaling plays a crucial role in osteoclastogenesis [38]. It has been reported that the NF-κB components p50 and p65 were recruited to the NFATc1 promoter upon RANKL stimulation and that the NF-κB inhibitor could suppress RANKL-induced osteoclastogenesis through downregulation of NFATc1 [39]. Thus, NF-κB plays an important role in the initial induction of NFATc1 during RANKL-induced osteoclastogenesis. Moreover, in this study, we showed that TAT-TN13 attenuated RANKL-induced NF-κB activation by performing luciferase reporter gene assay. Besides, Western blot analysis showed that TAT-TN13 suppressed phosphorylation of IκB and p65, the factors that leading to the up-regulation of NFATc1 expression. Suppressive effect of TAT-TN13 on a downstream target of RANKL could cause reduced activation of MAPK and consequently inhibit the expression of Cathepsin K andTRAP. Taken together, our data suggest that TN13 inhibits osteoclastogenesis via attenuating RANKL-induced p38, NF-κB and NFATc1 activation. However, a further study is required for a clear understanding of the inhibitory effect of TAT-TN13 on the regulatory mechanisms of p38, NF-κB and NFATc1 in osteoclastogenesis. Here, we demonstrated that TAT-TN13 protected mice against bone loss in the ovariectomy-based model of post-menopausal osteoporosis. The protective effects of TAT-TN13 against bone loss were determined by μCT analysis, H&E staining, and TRAP staining.

## 5. Conclusions

In conclusion, our results demonstrated the inhibitory effects of TAT-TN13 on osteoclastogenesis both in vitro and in vivo. Moreover, our study showed that TAT-TN13 mediated its effects through suppression of the p38, NF-κB and NFATc1 signaling pathways. Thus, our results suggest that TAT-TN13 may be a potential treatment drug for osteoclast-related diseases.

## Figures and Tables

**Figure 1 jcm-08-00431-f001:**
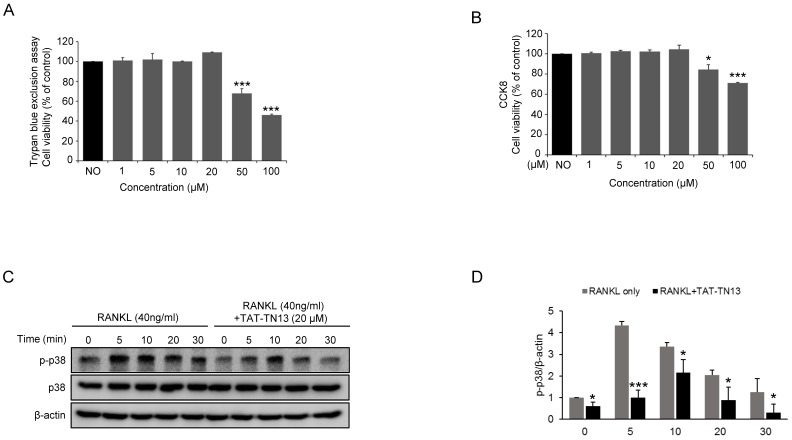
RANKL induces p38 MAPKs activation and TAT-TN13 inhibits it in a time-dependent manner without cytotoxic effects in vitro. Viability of TAT-TN13 treated RAW 264.7 cells was tested by a trypan blue assay (**A**) and CCK8 assays (**B**) at 24 h with various concentration. (**C**) RAW 264.7 cells were treated with or without 20 μM TAT-TN13 for 1 h and then treated with 40 ng/mL RANKL for the indicated times. Cell lysates were analysed by Western blotting with the anti-phospho-p38 antibody, anti-p38 antibody and *β*-actin (control). (**D**) Statistical analysis of p-p38 levels in (**C**). These experiments were repeated three times. The results are presented as the mean ± S.D of three independent experiments (* *p* < 0.05 and *** *p* < 0.001; n.s.: not significant, (*n* = 3)).

**Figure 2 jcm-08-00431-f002:**
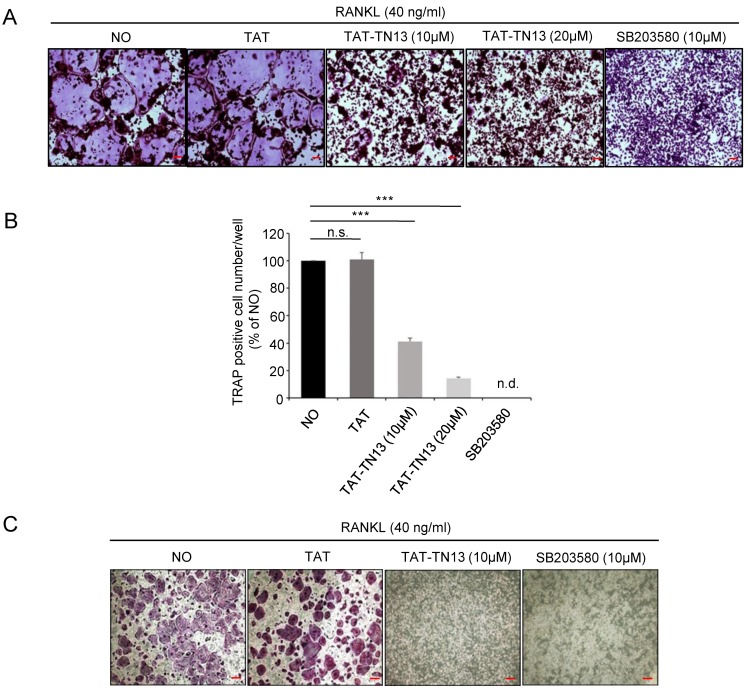
TAT-TN13 inhibits RANKL-induced osteoclast formation in a dose-dependent manner. (**A**) RAW 264.7 cells were incubated with RANKL (40 ng/mL) alone, or with plus various concentrations of TAT-TN13 and then stained for TRAP expression (scale bars: 200 µm). (**B**) The TRAP-positive osteoclasts generated in (**A**) were counted. In SB203580-treated cells, no osteoclast formation could be detected (n.d. indicates not detected). The results are presented as the mean ± S.D of three independent experiments (*** *p* < 0.001; n.s.: not significant). (**C**) BMMs were cultured for 5 days with M-CSF (30 ng/mL), RANKL (50 ng/mL) alone, or with plus TAT-TN13 (10 µM). The cells were fixed, subjected to TRAP staining (scale bars: 100 µm). These experiments were repeated three times. NO: control, which was treated with PBS; TAT: treated control, which was treated with control peptide.

**Figure 3 jcm-08-00431-f003:**
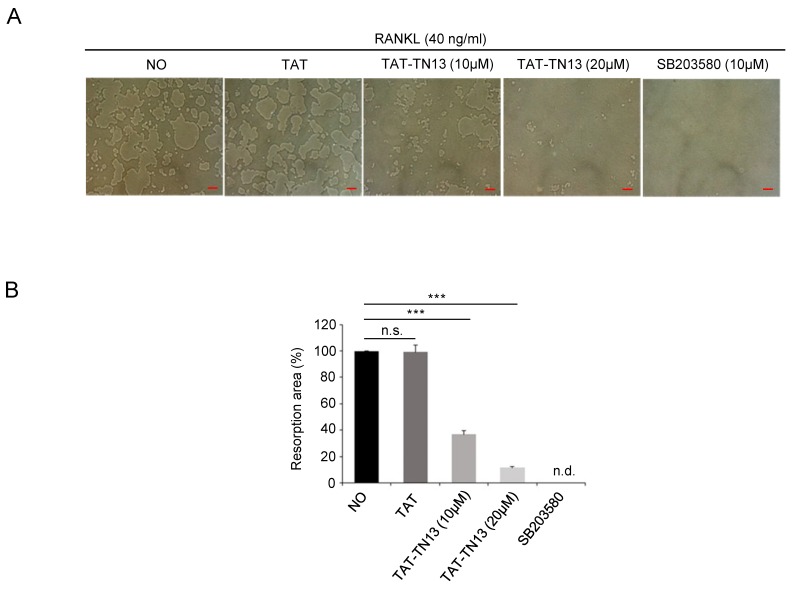
TAT-TN13- treated cells show reduced osteoclastic bone resorption. RAW 264.7 cells seeded into Corning Osteo Assay Surface plate and treated with different concentrations of TAT-TN13 (0, 10 and 20 μM) in the presence of RANKL (40 ng/mL) for 4 days. Cells attached to the plates were removed. (**A**) Resorption pits on the plates were captured using a light microscope (IX71; Olympus) (scale bars: 50 µm). (**B**) Resorption pit areas were quantified using Olympus CellSens imaging software. In SB203580-treated cells, no osteoclastic bone resorption could be detected (n.d. indicates not detected). The results are presented as the mean ± S.D of three independent experiments (*** *p* < 0.001; n.s.: not significant). NO: control, which was treated with PBS; TAT: treated control, which was treated with control peptide.

**Figure 4 jcm-08-00431-f004:**
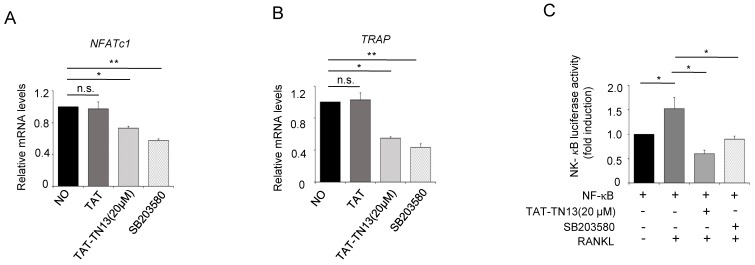
TAT-TN13 suppresses RANKL-induced osteoclast-specific gene expression and NF-κB activity. (**A**,**B**) RAW 264.7 cells were pretreated with different concentrations of TAT-TN13 and the p38 inhibitor, SB 203580, for 1 h and then treated with RANKL (40 ng/mL) for 3 h, 12 h. Gene expression was analyzed using real-time PCR (*n* = 3). (**C**) RAW 264.7 cells that were stably transfected with a NF-κB luciferase reporter construct were pretreated with varying concentrations of TAT-TN13 for 1 h and then treated with RANKL (40 ng/mL) for 8 h. Luciferase activity was measured using a dual-luciferase reporter assay system. The results are presented as the mean ± S.D of three independent experiments (* *p* < 0.05 and ** *p* < 0.01; n.s.: not significant). NO: control, which was treated with PBS; TAT: treated control, which was treated with control peptide.

**Figure 5 jcm-08-00431-f005:**
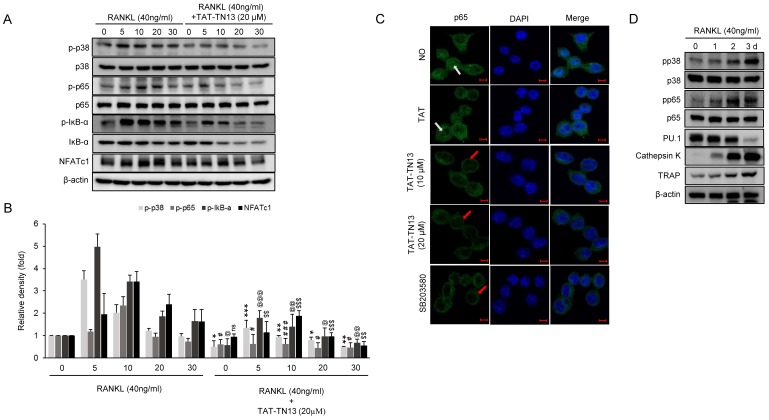
TAT-TN13 Inhibits NFATc1 activation by suppressing RANKL-induced phosphorylation of p65 in RAW 264.7 cells. (**A**) RAW 264.7 cells were pretreated in the presence or absence of TAT-TN13 and stimulated with RANKL (40 ng/mL) for the indicated times. Whole cell lysates were subjected to Western blot analysis with the indicated antibodies. (**B**) Statistical analysis of p-p38, p-p65, p-IκB-α, and NFATc1 levels in (**A**). These experiments were repeated three times. (**C**) RAW 264.7 cells were analyzed by immunofluorescence microscopy for nuclear localization of p65. The cells were pretreated for 1 h with or without 20 µΜ TAT-TN13 and then stimulated with RANKL for 30 min. The green color represents p65 staining, the blue color represents nuclei staining (scale bars: 5 µm). Red arrow indicates the localization of p65 in the cytoplasm, and white arrow specifies the presence of p65 in the nucleus. (**D**,**E**) Expression of osteoclast-specific genes during RANKL-mediated osteoclastogenesis. RAW 264.7 cells were cultured with RANKL without TAT-TN13 for the indicated periods. Cells were cultured with RANKL in the presence or absence of TAT-TN13 for 3 days. Cell lysates were analyzed using Western blotting. The expression of phospho-p38, p38, phospho-p65, p65, PU1, Cathepsin k, TRAP and β-actin was evaluated. (**F**) Statistical analysis of p-p38, p-p65, PU.1, Cathepsin K, and TRAP levels in (**E**). These experiments were repeated three times. NO: control, which was not treated; TAT: treated control, which was treated with control peptide. The results are presented as the mean ± S.D of three independent experiments (*^,#,@,$^
*p* < 0.05, **^,##,@@,$$^
*p* < 0.01, ***^,###,@@@,$$$^
*p* < 0.001; n.s.: not significant).

**Figure 6 jcm-08-00431-f006:**
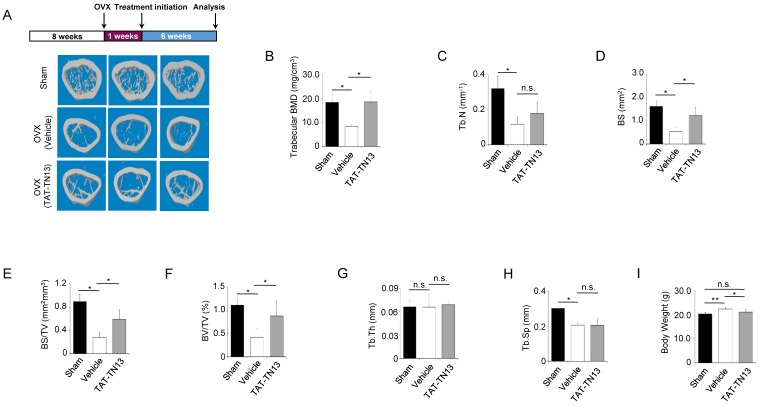
TAT-TN13 effectively prevents OVX-induced bone loss in vivo. (**A**) The fixed femurs were analysed by using micro-CT, and three-dimensional reconstructed images from each group are shown. (**B–H**) For each sample, bone mineral density (BMD), trabecular number (Tb.N), bone surface (BS), bone surface per total volume (BS/TV), bone volume per total volume (BV/TV), trabecular thickness (Tb.Th) and trabecular separation (Tb.Sp) were measured. (**I**) Body weight gain of OVX mice. The results are presented as the mean ± S.D of two independent experiments (* *p* < 0.05, ** *p* < 0.01 and *** *p* < 0.001; n.s.: not significant). (**J**,**K**) The femurs were fixed in 4%paraformaldehyde, decalcified, embedded, and sectioned as described in the Experimental section. Sections of femur were stained with H&E (magnification 10×, 20×, scale bars: 100 µm) and TRAP (magnification 10×, 40×, scale bars: 100 µm). Representative image was observed by using a microscope. These experiments were repeated 2 times (*n* = 6/group). Red arrows in (**J**) indicate bone loss and areas of osteolysis in the vehicle-treated OVX group. Green arrows indicate decreased osteoclast surface in the TAT-TN13-treated OVX group and red arrows indicate increased osteoclast surface in the vehicle-treated OVX group in (**J**).

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
