# Peer review of "Inhibition of Osteoclastogenesis by Thioredoxin-Interacting Protein-Derived Peptide (TN13)"

_jcm, 2019, doi:10.3390/jcm8040431_

Round 1
Reviewer 1 Report
In the present manuscript, authors examined the anti-resorptive effect of the p38 inhibitor TAT conjugated thioredoxin-interacting protein-derived peptide (TAT-TN13), both in vitro and in a pre-clinical mouse model of osteoporosis. The authors found that TAT-TN13 inhibits osteoclast differentiation and resorption activity and prevented bone loss in ovariectomized (OVX) mice by regulating the p38 and NF-κB signaling pathways.
Overall, the present study may provide proper scientific validation and significance to the potential of TAT-TN13 in the treatment or prevention of bone loss diseases.
However, I have some major remarks about the study design and the description of the experimental protocols.
Major comments:
1) Authors should indicate the number of biological and technical replicates for each assay performed
2) Please, provide some references supporting the use of the 10 uM concentration of SB203580 as positive control for p38 inhibition
3) Section 2.6: How authors explain the decision to add TAT-TN13 to the culture before cell differentiation to investigate the effect of the p38 inhibitor on osteoclast-mediated bone resorption? If the aim of this assay is to test the inhibition of osteoclast-mediated bone resorption by TAT-TN13, such p38 inhibitor should be probably add to mature osteoclasts. Thus, I suggest to design the experiment in order to allow Raw 264.7 to differentiate under RANKL stimulation until the formation of mature osteoclasts, and to add TAT-TN13 only after checking osteoclast differentiation by TRAP staining
4) Section 3.1: authors should test the cytotoxicity of TAT-TN13 for a longer period of time, i.e. 4 days, in order to match the timing required for osteoclast differentiation
5) Section 3.1: authors should calculate the EC50 value
6) Figure 2: please specify the legend in the figure caption: it is not so much clear what “NO” and “TAT” refer to
7) Lane 315: not all the signalling molecules involved in osteoclast differentiation and activation are increased in a time-dependent manner. Please, specify what molecules are increased.
Author Response
Reviewer-1
In the present manuscript, authors examined the anti-resorptive effect of the p38 inhibitor TAT conjugated thioredoxin-interacting protein-derived peptide (TAT-TN13), both in vitro and in a pre-clinical mouse model of osteoporosis. The authors found that TAT-TN13 inhibits osteoclast differentiation and resorption activity and prevented bone loss in ovariectomized (OVX) mice by regulating the p38 and NF-κB signaling pathways.
Overall, the present study may provide proper scientific validation and significance to the potential of TAT-TN13 in the treatment or prevention of bone loss diseases.
However, I have some major remarks about the study design and the description of the experimental protocols.
Response:
We appreciate the reviewer’s comments. Our manuscript has been edited by native English editors again. We have rearranged the manuscript as reviewers mentioned. Please check a new manuscript version and modified manuscript was clearly highlighted by “track changes” function in Microsoft Word.
Major comments:
Authors should indicate the number of biological and technical replicates for each assay performed
Response:
As reviewer mentioned, we have added the number of replicates in the figure legend.
Please, provide some references supporting the use of the 10 uM concentration of SB203580 as positive control for p38 inhibition
Response:
We have added two references (18, 19) in the 3.1 section.
Section 2.6: How authors explain the decision to add TAT-TN13 to the culture before cell differentiation to investigate the effect of the p38 inhibitor on osteoclast-mediated bone resorption? If the aim of this assay is to test the inhibition of osteoclast-mediated bone resorption by TAT-TN13, such p38 inhibitor should be probably add to mature osteoclasts. Thus, I suggest to design the experiment in order to allow Raw 264.7 to differentiate under RANKL stimulation until the formation of mature osteoclasts, and to add TAT-TN13 only after checking osteoclast differentiation by TRAP staining
Response:
The aim of this assay was only for determining the bone resorbing activity of mature osteoclasts. The differences may be due to the reduced numbers of mature osteoclasts. As reviewer mentioned, we performed additional experiments to determine whether TAT-TN13 affected the bone resorbing activity of osteoclasts. We fully differentiated RAW 264.7 cells into osteoclasts for 4 days and then treated TAT-TN13 or SB203580 for 12h. We found that TAT-TN13 or SB203580 treatment could not inhibit the bone resorbing activity of mature osteoclasts. We have described these results in the section 3.2 and added supplementary figures (Figure S2A, B).
Section 3.1: authors should test the cytotoxicity of TAT-TN13 for a longer period of time, i.e. 4 days, in order to match the timing required for osteoclast differentiation
Response:
We tested long-term cytotoxicity of TAT-TN13 with or without RANKL for 4 days in RAW 264.7 cells. Cell viability was not changed at doses below 20 μM of TAT-TN13 treatment. We have described these results in section 3.1 and added supplementary figures (Figure S1A, B).
Section 3.1: authors should calculate the EC50 value
Response:
50% lethal concentration (LC50) of TAT-TN13 in RAW264.7 cells was 86.04 µM at 24 h incubation. We have described it in section 3.1.
Figure 2: please specify the legend in the figure caption: it is not so much clear what “NO” and “TAT” refer to
Response:
We have added these information “NO: control, which was treated with PBS; TAT: treated control, which was treated with control peptide” in the figure legend.
Lane 315: not all the signalling molecules involved in osteoclast differentiation and activation are increased in a time-dependent manner. Please, specify what molecules are increased.
Response:
We have described in detail for the regulation of each proteins.

Reviewer 2 Report
The authors have analyzed the influence of TAT-TN13, a previously established inhibitor of p38a, on osteoclastogenic differentiation of RAW264.7 and primary bone marrow cells. They observed that TAT-TN13 inhibits RANKL-induced osteoclast differentiation, which was molecularly explained by impaired NF-kB signaling and Nfatc1 expression. They additionally provide in vivo evidence, i.e. protection against OVX-induced bone loss, to support the principal therapeutic impact of their findings. Although an impact of NF-kB signaling inhibition on osteoclastogenesis has been previously demonstrated, it is potentially relevant that TAT-TN13 might act more specifically than other inhibitors. On the other hand, since the authors fully focused on osteoclastogenesis inhibition, they can only speculate about the reduced side effects mediated by TAT-TN13 treatment.
Specific comments:
1) Although the overall data are truly convincing, it is required to provide quantitative data for all Western blots shown throughout the manuscript.
2) The conclusion that TAT-TN13 reduced the bone resorbing activity of mature osteoclasts is not supported by the experiment shown in Figure 3. In fact, since, according to the Method section, TAT-TN13 was present during the whole course of differentiation, the near absence of resorption pits is probably due to the reduced number of osteoclasts. In order to support the above-mentioned conclusion, the authors would have to add TAT-TN13 to the cultures at a time, where osteoclasts have already been generated.
3) The images shown in Figure 5B are too small to allow a full appreciation of the differences that were observed. The same applies for Figure 6I/J. Here it would also help to highlight osteoclasts by arrows.
4) One key argument of the authors is that TAT-TN13, based on its higher specificity for p38a, may have less side effects compared to other anti-resorptive agents. Importantly however, they only analyzed the skeletal phenotype of the treated mice and did not check for adverse events in other organs. While this might be difficult to perform retrospectively, the authors should at least provide the body weight (and possible few other baseline parameters) for the mice shown in Figure 6.
Author Response
Reviewer-2
The authors have analyzed the influence of TAT-TN13, a previously established inhibitor of p38a, on osteoclastogenic differentiation of RAW264.7 and primary bone marrow cells. They observed that TAT-TN13 inhibits RANKL-induced osteoclast differentiation, which was molecularly explained by impaired NF-kB signaling and Nfatc1 expression. They additionally provide in vivo evidence, i.e. protection against OVX-induced bone loss, to support the principal therapeutic impact of their findings. Although an impact of NF-kB signaling inhibition on osteoclastogenesis has been previously demonstrated, it is potentially relevant that TAT-TN13 might act more specifically than other inhibitors. On the other hand, since the authors fully focused on osteoclastogenesis inhibition, they can only speculate about the reduced side effects mediated by TAT-TN13 treatment.
Response:
We appreciate the reviewer’s comments. Our manuscript has been edited by native English editors again. We have rearranged the manuscript as reviewers mentioned. Please check a new manuscript version and modified manuscript was clearly highlighted by “track changes” function in Microsoft Word.
Specific comments:
Although the overall data are truly convincing, it is required to provide quantitative data for all Western blots shown throughout the manuscript.
Response:
We have added statistical analysis of Western blots in the figures.
The conclusion that TAT-TN13 reduced the bone resorbing activity of mature osteoclasts is not supported by the experiment shown in Figure 3. In fact, since, according to the Method section, TAT-TN13 was present during the whole course of differentiation, the near absence of resorption pits is probably due to the reduced number of osteoclasts. In order to support the above-mentioned conclusion, the authors would have to add TAT-TN13 to the cultures at a time, where osteoclasts have already been generated.
Response:
We agree with your comments. Reviewer1, and 3 also have raised same comments. To address this question, we performed additional experiments to determine whether TAT-TN13 affected the bone resorbing activity of osteoclasts. We fully differentiated RAW 264.7 cells into osteoclasts for 4 days and then treated TAT-TN13 or SB203580 for 12h. We found that TAT-TN13 or SB203580 treatment could not inhibit the bone resorbing activity of mature osteoclasts. We have described these results in the section 3.2 and added supplementary figures (Figure S2A, B).
The images shown in Figure 5B are too small to allow a full appreciation of the differences that were observed. The same applies for Figure 6I/J. Here it would also help to highlight osteoclasts by arrows.
Response:
We highlighted each figures with colored arrows.
One key argument of the authors is that TAT-TN13, based on its higher specificity for p38a, may have less side effects compared to other anti-resorptive agents. Importantly however, they only analyzed the skeletal phenotype of the treated mice and did not check for adverse events in other organs. While this might be difficult to perform retrospectively, the authors should at least provide the body weight (and possible few other baseline parameters) for the mice shown in Figure 6.
Response:
We have added the body weight data in Figure 6I, TAT-TN13 blocked body weight gain following ovariectomy and did not show any severe phenotypes in the mice.

Reviewer 3 Report
The manuscript by Kim et al describes the effect of TAT-conjugated TN13 on osteoclast differentiation and activity in vitro, and in bone microarchitecture in vivo. The authors conclude that the compound by inhibiting osteoclastogenesis through its actions on the p38 MAPK and NFkB pathways, could be useful for preventing or treating osteoporosis. This is an interesting possibility. However, the study is preliminary and the authors did not take advantage of the in vivo study to fully understand the actions and mechanism of TAT-TN13. In addition, the manuscript should be revised for the use of the English language, to avoid the use of incorrect words and grammatical errors.
Specific issues:
1- In the introduction, the authors indicate that TAT-TN13 could be used as treatment for osteoclast-related diseases. This statement is incorrect, since the compound should not be used for the treatment of osteopetrosis, a disease that results from the inability of osteoclast to resorb bone. It would be more appropriate to indicate that the compound could be used to treat diseases with “exacerbated osteoclast formation or activity”.
2- Methods:
a. Cell density should be indicated /cm2 and not /well
b. Is the ratio cell suspension/Trypan blue correct? Trypan blue is usually diluted to 0.04% to estimate the proportion of dead cells
3- All western blots should be quantified, and the data should be reported as mean±SD in order to be able to infer changes in protein levels due to the treatments. For example, the authors indicate that p38MAPK phosphorylation was markedly suppressed by TAT-TN13 treatment, but this is not so evident in the images.
4- Figure 1A: the effect of the compound on cell viability by Trypan blue should be reported as % viable cells, rather than as number of cells. Since the compound might affect both viability and proliferation, cell number does not provide evidence of changes in viability.
5- Figure 1B: what is the control the cells were exposed to, used as 100% in this graph?
6- Undifferentiated Raw264.7 cells might have different sensitivity to the toxic effects of TAT-TN13 than differentiating osteoclasts. Therefore, the study reported in Figure 1 should be repeated under the same conditions as the experiments shown in subsequent figures (in the presence of RANKL/MCSF).
7- The results of the resorption activity assay parallels exactly the number of osteoclasts for each treatment condition. Therefore, it is possible that osteoclast do not have reduced resorption capability in the presence of TAT-TN13, and there is simply less of them to resorb the bone. This possibility should be properly discussed.
8- In order to determine whether TAT-TN13 indeed had an effect on osteoclast number and osteoclastic bone resorption, authors should perform proper histomorphometry analysis, counting osteoclast number and surface and eroded surface, all corrected by bone surface. Importantly, the levels of markers of bone resorption should be measured in the circulation. Otherwise, it is not possible to conclude that the compound reduces osteoclast activity in vivo.
Author Response
Reviewer-3
The manuscript by Kim et al describes the effect of TAT-conjugated TN13 on osteoclast differentiation and activity in vitro, and in bone microarchitecture in vivo. The authors conclude that the compound by inhibiting osteoclastogenesis through its actions on the p38 MAPK and NFkB pathways, could be useful for preventing or treating osteoporosis. This is an interesting possibility. However, the study is preliminary and the authors did not take advantage of the in vivo study to fully understand the actions and mechanism of TAT-TN13. In addition, the manuscript should be revised for the use of the English language, to avoid the use of incorrect words and grammatical errors.
Response:
We appreciate the reviewer’s comments. Our manuscript has been edited by native English editors again. We have rearranged the manuscript as reviewers mentioned. Please check a new manuscript version and modified manuscript was clearly highlighted by “track changes” function in Microsoft Word.
Specific issues:
In the introduction, the authors indicate that TAT-TN13 could be used as treatment for osteoclast-related diseases. This statement is incorrect, since the compound should not be used for the treatment of osteopetrosis, a disease that results from the inability of osteoclast to resorb bone. It would be more appropriate to indicate that the compound could be used to treat diseases with “exacerbated osteoclast formation or activity”.
Response:
We have added your comment “exacerbated osteoclast formation or activity” in the end of Introduction section.
2- Methods:
Cell density should be indicated /cm2 and not /well
Response:
We have corrected “well” to “cm2”.
Is the ratio cell suspension/Trypan blue correct? Trypan blue is usually diluted to 0.04% to estimate the proportion of dead cells
Response:
That was mislabeled. We have corrected the volume of Trypan blue (10 µL). We added 10 µL of trypan blue stock solution (0.4%) to 10 µL of cells. Finally, we used 0.2% mix.
All western blots should be quantified, and the data should be reported as mean±SD in order to be able to infer changes in protein levels due to the treatments. For example, the authors indicate that p38MAPK phosphorylation was markedly suppressed by TAT-TN13 treatment, but this is not so evident in the images.
Response:
Reviewer-2 also have raised same comment. We have added statistical analysis of Western blots in the figures.
Figure 1A: the effect of the compound on cell viability by Trypan blue should be reported as % viable cells, rather than as number of cells. Since the compound might affect both viability and proliferation, cell number does not provide evidence of changes in viability.
Response:
We have presented Figure 1A as % viable cells and additionally, we tested long-term cytotoxicity of TAT-TN13 with or without RANKL for 4 days in RAW 264.7 cells. Cell viability was not changed at doses below 20 μM of TAT-TN13 treatment. We have described these results in section 3.1 and added supplementary figures (Figure S1A, B).
Figure 1B: what is the control the cells were exposed to, used as 100% in this graph?
Response:
We have changed the labelling “0” to “NO”, control, which was treated with PBS.
Undifferentiated Raw264.7 cells might have different sensitivity to the toxic effects of TAT-TN13 than differentiating osteoclasts. Therefore, the study reported in Figure 1 should be repeated under the same conditions as the experiments shown in subsequent figures (in the presence of RANKL/MCSF).
Response:
We tested long-term cytotoxicity of TAT-TN13 with or without RANKL for 4 days in RAW 264.7 cells. Cell viability was not changed at doses below 20 μM of TAT-TN13 treatment. We have described these results in section 3.1 and added supplementary figures (Figure S1A, B).
The results of the resorption activity assay parallels exactly the number of osteoclasts for each treatment condition. Therefore, it is possible that osteoclast do not have reduced resorption capability in the presence of TAT-TN13, and there is simply less of them to resorb the bone. This possibility should be properly discussed.
Response:
Other reviewers also have raised same comments. To address this question, we performed additional experiments to determine whether TAT-TN13 affected the bone resorbing activity of osteoclasts. We fully differentiated RAW 264.7 cells into osteoclasts for 4 days and then treated TAT-TN13 or SB203580 for 12h. We found that TAT-TN13 or SB203580 treatment could not inhibit the bone resorbing activity of mature osteoclasts. We have described these results in the section 3.2 and added supplementary figures (Figure S2A, B).
In order to determine whether TAT-TN13 indeed had an effect on osteoclast number and osteoclastic bone resorption, authors should perform proper histomorphometry analysis, counting osteoclast number and surface and eroded surface, all corrected by bone surface. Importantly, the levels of markers of bone resorption should be measured in the circulation. Otherwise, it is not possible to conclude that the compound reduces osteoclast activity in vivo.
Response:
From additional experiments in Figure S2A, B, we could find that the reduced resorbing activity of osteoclasts was due to the reduced number of mature osteoclasts. Thus, we have limited the description in conclusion as “our results demonstrated the inhibitory effects of TAT-TN13 on osteoclastogenesis both in vitro and in vivo”. Li and colleagues (Endocrinology, 143:3105-3113, 2002) have also suggested that p38 MAPK-mediated signals are required for inducing osteoclast differentiation but not for osteoclast function. In Figure 6, TRAP staining and histomorphometric analysis of sections of femurs revealed that there was substantial reduction in both osteolysis and osteoclast surface in bone samples of TAT-TN13-treated OVX group in comparison with vehicle-treated OVX group. We have noted them by indicating with arrows.

Round 2
Reviewer 1 Report
The authors fully answered my questions. I have only one remark regarding cell viability after 4 days treatment with RANKL (40 ng/ml) and TAT-TN13 (10 or 20 µM) (Supplementary figure S1, panel B). Authors stated “cell viability was not changed at doses below 20 μM of TAT-TN13 treatment (Figure S1A, B)”. This is correct for the TAT-TN13 treatment alone. However, cell viability after 4 days treatment with TAT-TN13 (10 or 20 µM) with the addition of RANKL (40 ng/ml) was significantly impaired in respect to the control (NO) (Supplementary Figure S1B). How do authors explain such result? It is likely RANKL exerts a long-term cytotoxicity on Raw 264.7. In light of this result, I think that the Figure S1B should be deleted from the sentence “cell viability was not changed at doses below 20 μM of TAT-TN13 treatment (Figures S1A, B)”, keeping only Figure S1A, which referred to treatment with TAT-TN13 alone.
Author Response
The authors fully answered my questions. I have only one remark regarding cell viability after 4 days treatment with RANKL (40 ng/ml) and TAT-TN13 (10 or 20 µM) (Supplementary figure S1, panel B). Authors stated “cell viability was not changed at doses below 20 μM of TAT-TN13 treatment (Figure S1A, B)”. This is correct for the TAT-TN13 treatment alone. However, cell viability after 4 days treatment with TAT-TN13 (10 or 20 µM) with the addition of RANKL (40 ng/ml) was significantly impaired in respect to the control (NO) (Supplementary Figure S1B). How do authors explain such result? It is likely RANKL exerts a long-term cytotoxicity on Raw 264.7. In light of this result, I think that the Figure S1B should be deleted from the sentence “cell viability was not changed at doses below 20 μM of TAT-TN13 treatment (Figures S1A, B)”, keeping only Figure S1A, which referred to treatment with TAT-TN13 alone.
Response:
We agree with your comments. Because reviewer-3 has raised this comment to test TAT-TN13 cytotoxicity in the presence of RANKL, we have tried to test its cytotoxicity under RANKL treatment condition. As you know, CCK8 activity is dependent on the number of viable cells. Raw264.7 cells could proliferate under control (NO) treatment condition but they could not proliferate under RANKL treatment condition because they might be in differentiation process by RANKL treatment. It seemed that TAT-TN13 did not induce cell death under RANKL treatment condition. We have deleted Figure S1B from the sentence and Figure S1.

Reviewer 2 Report
The authors have responded to all my comments and singifcantly improved their manuscript.
Author Response
The authors have responded to all my comments and significantly improved their manuscript.
Response:
Thank you very much for reviewing our manuscript.
